# Another Move towards Bicalutamide Dissolution and Permeability Improvement with Acetylated β-Cyclodextrin Solid Dispersion

**DOI:** 10.3390/pharmaceutics14071472

**Published:** 2022-07-15

**Authors:** Tatyana V. Volkova, Olga R. Simonova, German L. Perlovich

**Affiliations:** G. A. Krestov Institute of Solution Chemistry, The Russian Academy of Sciences, 153045 Ivanovo, Russia; vtv@isc-ras.ru (T.V.V.); ors@isc-ras.ru (O.R.S.)

**Keywords:** β-cyclodextrin derivatives, complexation, supersaturation, dissolution, permeability

## Abstract

The complex formation of antiandrogen bicalutamide (BCL) with methylated (Me-β-CD) and acetylated (Ac-β-CD) β-cyclodextrins was investigated in buffer solution pH 6.8. A two-fold strongly binding of BCL to Ac-β-CD as compared to Me-β-CD was revealed. The solid dispersion of BCL with Ac-β-CD was prepared by the mechanical grinding procedure to obtain the complex in the solid state. The BCL/Ac-β-CD complex was characterized by DSC, XPRD, FTIR, and SEM techniques. The effect of Ac-β-CD in the BCL solid dispersions on the non-sink dissolution/permeation simultaneous processes was disclosed using the side-by-side diffusion cell with the help of the cellulose membrane. The elevated dissolution of the ground complex, as compared to the raw drug as well as the simple physical mixture, accompanied by the supersaturation was revealed. Two biopolymers—polyvinylpyrrolidone (PVP, M_n_ = 58,000) and hydroxypropylmethylcellulose (HPMC, M_n_ ~ 10,000)—were examined as the precipitation inhibitors and were shown to be useful in prolonging the supersaturation state. The BCL/Ac-β-CD complex has the fastest dissolution rate in the presence of HPMC. The maximal concentration of the complex was achieved at a time of 20, 30, and 90 min in the pure buffer, with PVP and with HPMC, respectively. The effectiveness of the BCL dissolution (release) processes (illustrated by the $AUC_{C(t)}$ parameter) was estimated to be 7.8-, 5.8-, 3.0-, and 1.8-fold higher for BCL/Ac-β-CD (HPMC), BCL/Ac-β-CD (PVP), BCL/Ac-β-CD (buffer), and the BCL/Ac-β-CD physical mixture, respectively, as compared to the BCL_raw sample. The excipient gain factor (EGF), calculated for the dissolution of the BCL complex, was shown to be 2.6 in the presence of HPMC, which is 1.3-fold greater as compared to PVP. From the experimental dissolution results, it can be concluded that the formation of BCL ground complex with Ac-β-CD enhances the dissolution rate of the compound. The permeation was also shown to be advantageous in the presence of the polymers, which was demonstrated by the elevated fluxes of BCL through the membrane. The comparison of the dissolution/permeation processes was illustrated and discussed. The conclusion was made that the presence of HPMC as a stabilizer of the supersaturation state is promising and seems to be a useful tool for the optimization of BCL pharmaceutical formulations manufacturing.

## 1. Introduction

Bicalutamide (N-[4-Cyano-3-(trifluoromethyl)phenyl]-3-[(4-fluorophenyl)sulfonyl]-2-hydroxy-2-methylpropanamide) (BCL) is a non-steroidal antitumoral antiandrogenic drug. By binding to androgen receptors on the surface of target cells, it makes them inaccessible to androgens while increasing the concentration of hormones in the blood plasma [1]. In spite of the fact that bicalutamide was patented in 1982 and approved for medical use in 1995 [2], it is now sold in many countries, including the most developed ones [3]. Due to this fact, new information on BCL’s biologically important properties, as well as the technologies aimed at the improvement of these properties, seems to be useful.

Bicalutamide (BCL) (Figure 1) joins the large BCS class II drugs characterized by very poor solubility and low dissolution rate, which is crucial for gastrointestinal absorption and, as follows, oral bioavailability. The tablets of BCL (trade name CASODEX^®®)^) contain 50 mg of BCL taken daily.

The extremely poor aqueous solubility of bicalutamide has been recognized in the literature as being approximately 8.85 mg·L^−1^ [4], 3.7 μg·mL^−1^ [5], and less than 5 mg∙L^−1^ [6]. In addition, in our previous studies, it was estimated to be 8.1 × 10^−6^ M, 1.30 × 10^−5^ M, and 1.64 × 10^−5^ M in water (25 °C, 35 °C, and 40 °C, respectively) [7,8], 1.44 × 10^−5^ M (buffer pH 7.4, 25 °C) [9]. The *pK_a_* value of BCL equal to 11.49 (calculated by ACD/LABS) implies that the solubility in aqueous media is independent of pH in the biologically relevant region. In order to improve an insufficient aqueous solubility, an appropriate form of the drug substance or its special formulation need to be developed. A whole number of approaches aimed at improving the poor solubility/dissolution rate exist to date. For example, salts, polymorphs, solvates, hydrates, and co-crystals production [10,11,12,13]. In addition to this, drug delivery systems involving different pharmaceutically friendly excipients serving as solubilizing and stabilizing agents have not lost actuality from the 19th century to nowadays. Among the excipients, cyclic oligosaccharides—cyclodextrins (CD)—represent a huge share since they can interact with drug molecules, forming inclusion complexes [14]. The ability of CDs to form inclusion complexes can be used in drug formulation to improve the solubility of poorly water-soluble class II drugs. Since the inclusion complexes are formed by the molecular encapsulation of a drug into the hydrophobic cavity of the CD by means of the interactions between the guest and the host, evidently, the efficacy of the solubilization depends both on the specific features of the drug and the nature of cyclodextrin [15]. Moreover, since large amounts of CDs in drug formulations should be avoided [14], it is preferential to use the cyclodextrins with the highest solubilization potential. In this respect, the substituted CD (hydroxypropylated, methylated, acetylated, etc.) derivatives often appear to have an advantage compared to “parent” ones [16]. Sometimes, methylated β-CDs are more effective than parent β-CD and even HP-β-CD [17]. Moreover, acetylating the hydroxyl groups at the glucose units of heptakis (2,6-di-O-methyl)-β-cyclodextrin was shown to be successful in reducing the hemolytic activity and muscular tissue irritation, thus enhancing the bioadaptability [18].

In the present study, attempts to improve the solubility and dissolution rate of BCL using methylated β-CD and acetylated β-CD were undertaken. Notably, there are a number of papers on BCL solubility enhancement both via biopolymers, such as hydroxypropyl methylcellulose [19,20], lactose [21], polyvinylpyrrolidone [22,23,24], poly (ethyleneoxide) [25], and surfactants, such as sodium lauryl sulfate [26], poloxamers [27,28], and cyclodextrins [4,29,30]. An analysis of the available literature sources showed that the BCL solubility was improved 2-fold by β-cyclodextrin inclusion complexation (*K_C_* = 161.36 M^−1^) [4], and the dissolution rate can be enhanced—more than 95% drug was released in 10 min [29]. As it was revealed from the phase solubility in the study of Patil et al. [30], the aqueous solubility of bicalutamide was increased by 91% in a binary complex with hydroxypropyl-β-cyclodextrin and the time required to release 90% BCL from inclusion complex was 15 min (the release from pure BCL was incomplete even in 60 min). An analysis of the BCL spectral characteristics during the BCL/β-cyclodextrin complex formation revealed a red shift of absorption *λ_max_* from 270.0 nm (in water) to 274 nm (in 10 × 10^−3^ M β-CD) [31] due to the transition to less protic environments.

It should be emphasized that a majority of the studies are devoted to experiments with BCL solid complexes. At the same time, in our opinion, disclosing the nature of the interactions of a drug compound with CD molecules in solutions is no less important. Determining the complex stoichiometry and solubilizing potential of different CDs for the drug enables the selection of the most effective solubilizing agent for successful pharmaceutical forms. Moreover, for the first time, the thermodynamics of the bicalutamide complex formation and, in so doing, the driving forces of the processes and the factors controlling the supramolecular interactions were disclosed from the experiments in solutions.

The permeation of the drug through the biological barriers is no less significant than an adequate solubility/dissolution rate. All of these processes determine the drug bioavailability and transport to the receptors on the target organ [32]. The addition of cyclodextrins to solubilize drugs could also alter their permeability behavior. Similar to the solubility, in the presence of the excipients, ex., CDs, the permeability depends on both the nature of the drug and the structural features of cyclodextrin. As it was underlined in Sigurdsson et al. [33], cyclodextrins will improve the permeation rate through a diffusion-controlled layer but hinder the diffusion through a lipophilic membrane-controlled barrier. For fast evaluation of the CD impact on the permeation flux and permeability coefficient without using biological tissues, different types of artificial barriers are examined. Among them, semi-permeable membranes from the regenerated cellulose proved to be advantageous [34]. For the novelty, the present study evaluated the dissolution and permeation processes of bicalutamide simultaneously. It seemed reasonable since, in the case of different formulations (ex. based on cyclodextrins, polymers, micelles, and co-crystals), the combined dissolution/permeation approaches are based on cell models [35] and, at a later date, on artificial membranes [36,37] are limited. As was shown by Buckley et al. in the critical review of the experimental data available in the literature pertaining to the solubility and permeability of poorly soluble drugs from enabling formulations [38], the simultaneous dissolution and permeation examination is extremely important in the case of co-solvents and solubilizing additives often leading to a supersaturated state of the drug in a delivery system. For the best, the permeation flux instead of the permeability coefficient (flux to the drug concentration in the donor solution ratio) can be taken and discussed together with the dissolution process. As was stated by Sironi et al. [39], combining dissolution testing with permeation studies facilitates the time- and cost-effectiveness of the early-stage evaluation of absorption-enabling formulations. In the case of the supersaturation state often occurring upon the dissolution of the drug formulation based on, for example, amorphous solid dispersions, the best precipitation inhibitors should be applied in order to optimize oral bioavailability [40]. Effective inhibition always results in the highest oral bioavailability, as was proved by Vandecruys et al. [41].

In the present study, we focused on the investigation of the BCL solubility, dissolution rate, and permeability in the presence of methylated β-CD (Me-β-CD) and acetylated β-CD (Ac-β-CD). The experiments in solutions were carried out with both CDs. The solubilizing potential of CDs towards BCL and the apparent stability constants were estimated from the phase solubility diagrams constructed at several temperatures from 298.15 K to 313.15 K. The thermodynamic functions of the complex formation were determined. The influence of the CD structure on these parameters was estimated. Thermodynamic parameters were taken into the discussion for a comparison of the driving forces of the outlined processes.

The second part of the study was devoted to the preparation of the solid BCL/Ac-β-CD complex by a grinding procedure. The complex was characterized by differential scanning calorimetry (DSC), X-ray powder difractometry (PXRD), IR-spectroscopy, and scanning electron microscopy (SEM). The impact of the complex formation on the solubility, dissolution profile, sink and non-sink permeation using a vertical-type Franz diffusion cell and a side-by-side (7 mL/7 mL) diffusion cell, respectively, and an artificial membrane MWCO 12–14 kDa was disclosed and discussed. In order to delay the supersaturation state upon the dissolution of the BCL/Ac-β-CD complex, two biopolymers (polyvinylpyrrolidone and hydroxypropylmethylcellulose) were successfully applied.

## 2. Materials and Methods

### 2.1. Materials

Bicalutamide (purity ≥ 98%) was obtained from a commercial source (Merck, KGaA Darmstadt, Germany). Methylated β-CD (Me-β-CD) (M_n_ = 1345.0 g·mol^−1^, DS-12, purity > 95%) and acetylated β-CD (Ac-β-CD) (M_n_ = 1765.0 g·mol^−1^, DS-7, purity > 95%) were purchased from CycloLab.LTD (Budapest, Hungary). Polyvinylpyrrolidone K29-32 (PVP, M_n_ = 58,000) and hydroxypropylmethylcellulose (HPMC, M_n_ ~ 10,000) were obtained from Sigma-Aldrich (St. Louis, MO, USA). Double distilled water with an electrical conductivity of 2.1 μS cm^−1^ (PWT H198308, HANNA^®^ instruments) was used for the preparation of the solutions. Potassium dihydrogen phosphate (purity ≥ 99%) and sodium hydroxide (purity ≥ 98%) were obtained from Merk (Darmstadt, Germany). A buffer solution with pH 6.8, was prepared in the following way: 27.22 g of KH_2_PO_4_ was dissolved in 1 L of H_2_O (solution 1), and 2 g of NaOH was dissolved in 250 mL of H_2_O (solution 2); 250 mL and 112 mL of solutions 1 and 2, respectively, were combined and reduced to 1 L. The pH value was measured by using a pH meter (Five Go^TM^ F2, Mettler Toledo, (Greifenzee, Switzerland)) standardized with pH 4 and pH 7 solutions.

### 2.2. Methods

#### 2.2.1. UV-Spectroscopy

The UV-spectra of pure BCL and BCL/Me-β-CD, BCL/Ac-β-CD complexes in a buffered solution were obtained with the help of a spectrophotometer (Cary 50 spectrophotometer Varian, Palo Alto, CA, USA, Software Version 3.00 (339)). The solutions were scanned in the region from 200 to 500 nm. The appropriate dilution of the BCL/CD solutions did not cause changes to the spectra shape and the position of the maximum.

#### 2.2.2. Determination of BCL Phase Solubility in Me- and Ac-β-CD Solutions

The shake-flask procedure, first developed by Higuchi and Connors [42], was used in order to construct the phase-solubility diagrams. To this end, we determined the BCL concentrations in the saturated solutions of pure pH 6.8 buffer and at different Me- and Ac- β-CD concentrations (0.0025, 0.005, 0.01, and 0.015 M). Glass vials containing an excess amount of BCL in a CD solution of a specific concentration were placed in an air thermostat at 298.15, 303.15, 308.15, and 313.15 K (±0.05) K and mixed for 72 h. The time of 72 h was estimated from the kinetic study as sufficient for the thermodynamic equilibrium. The suspensions were kept out overnight and then centrifuged under at a predetermined temperature for 20 min at 12,000 rpm (Biofuge pico, Thermo Electron LED GmbH, Langenselbold, Germany). Aliquots of the saturated solutions were taken with a Proline^®®^ pipette (Sartorius Biohit Liquid Handling Oy Laippatie 1.00880 Helsinki, Finland), diluted if necessary, and the absorbance was measured with the help of a spectrophotometer (Cary 50 spectrophotometer Varian, Palo Alto, CA, USA, Software Version 3.00 (339)) with an accuracy of 2–4%. The absorbance was recalculated to the concentration using the calibration curve (calibration range 6 × 10^−6^ ÷ 2 × 10^−6^ M bicalutamide concentrations). The solubility values are reported as an average of at least three replicated experiments.

#### 2.2.3. Determination of the Complex Stoichiometry with Job’s Plot

The stoichiometry of the BCL/CDs inclusion complexes was examined via a continuous variation technique developed by Job [43]. To this end, the equimolar solutions of BCL and Ac-β-CD of 8.14 × 10^−6^ and 8.49 × 10^−6^ M concentrations in the buffer solution pH 6.8 were mixed to a specific volume, varying the molar ratio but keeping the total concentration constant. The absorbance (*A*) at *λ_max_* was measured for all of the solutions, and the difference in absorbance (Δ*A*) in the presence and in the absence of CD was plotted as a function of *R* = *C*_BCL_/(*C*_BCL_ + *C*_Ac-β-CD_).

#### 2.2.4. Preparation of the Solid Samples by Mechanical Grinding Procedure

The physical mixture of the two components was prepared by mixing parent BCL with Ac-β-CD in a 1:1 ratio by the use of a spatula. The ground samples of the BCL and BCL/Ac-β-CD complex were processed from the parent BCL and BCL/Ac-β-CD physical mixture (solid samples). The sample was put into an agate jar (12 mL) with milling balls (agate, 5 mm) and placed in a Planetary ball micro mill Fritsch Pulverisette 7 (Idar-Oberstein, Germany) for 1 h with a pause period to avoid mechanical heating. The rotational speed was 600 rpm. The resulting products were collected and subjected to DSC, PXRD, IR, and SEM characterization. The samples for the dissolution/permeation examination were additionally sieved through a 150 mm mesh and stored in a desiccator with sunlight protection.

#### 2.2.5. Differential Scanning Calorimetry

The DSC profiles of the samples were obtained with the help of a differential scanning calorimeter (Perkin Elmer DSC 4000, Perkin-Elmer Analytical Instruments, Norwalk, CT, USA) with a refrigerated cooling system (Norwalk, CT, USA). The samples were heated in standard aluminum sample holders. The heating rate of 10 K min^−1^ was applied. The experiments were carried out in a nitrogen atmosphere. The unit was calibrated with indium and zinc standards. The accuracy of the weighing procedure was ±0.01 mg.

#### 2.2.6. PXRD Analysis

The powder XRD data of the bulk materials were recorded under ambient conditions on a D2 Phaser Bragg-Brentano diffractometer (Bruker AXS, Karlsruhe, Germany) with a copper X-ray source (λ_CuKα1_ = 1.5406 Å) and a high-resolution position-sensitive LYNXEYE XE-T detector. The quantity of the sample used for the PXRD analysis was 50 mg. The samples were placed into the plate sample holders and rotated at a speed of 15 rpm during the data acquisition.

#### 2.2.7. IR-Spectroscopy

The Fourier-transform infrared spectra of the BCL, Ac-β-CD, BCL/Ac-β-CD physical mixture, and solid complex were recorded with the help of an FTIR spectrometer, VERTEX 80 v. All of the samples were pressed into KBr pellets, and the spectra were scanned over a frequency range of 4000–350 cm^−1^.

#### 2.2.8. Scanning Electron Microscopy

Scanning electron microscopy, using a high-performance microscope (Quattro S, Thermo Fisher Scientific, Brno, Czech Republic), was applied for the examination of the surface morphologies of the BCL/Ac-β-CD solid complex. The microscopic structures of the parent BCL, as well as the ground BCL, Ac-β-CD, and BCL/Ac-β-CD physical mixture were also obtained for comparison. The acceleration voltage was equal to 20–30 kV, and the magnification (mag) of the micrographs were (×400), (×1300), and (×5000) for all of the samples.

#### 2.2.9. Dissolution/Permeation Setup in a Side-by-Side Cell

The simultaneous evaluation of the dissolution and permeation performances was carried out in a side-by-side cell (PermeaGear.de H1C SIDE-BI-SIDE Diffusion System, SESGmbH-Analytical system, Bechenheim, Germany). The membrane of the regenerated cellulose MWCO 12–14 kDa (Visking dialysis tubing MWCO 12–14 kDa, Medicell Membranes Ltd., London, UK) was applied as a barrier between the donor and receptor cells. A comprehensive review of using the cellulose membranes for drug/CD formulations was reported by Loftsson et al. [44]. The volume of the donor/acceptor compartment was 7 mL. An effective permeation area was 1.77 cm^2^. Taking a single oral dose of BCL of 50 mg and a volume in the intestinal lumen of 250 mL into consideration, the powdered sample of 1.7 mg BCL or an equivalent amount of the BCL/Ac-β-CD physical mixture or solid complex was placed in the donor cell; the cells were assembled with a membrane between them. The receptor compartment was filled with 7 mL of buffer pH 6.8, then an equal buffer volume was added to the donor compartment (start of the experiment). In the case of the samples with Ac-β-CD, the receptor solution contained the amount of CD equivalent to that of the solid sample. The fixed 0.01% (2.5 g·L^−1^) and 0.25% (0.1 g·L^−1^) concentrations of PVP and HPMC, respectively, in pH 6.8 buffer were applied in both donor and acceptor compartments for the experiments in the presence of these excipients. The solutions in both cells were stirred with stirring bars at a fixed speed of 500 rpm. The cells were thermostated at 37 °C during the experiment via water circulating. The samples of the solutions from the donor (*V* = 0.5 mL) and receptor (*V* = 0.4 mL) cells were withdrawn every 30 min and immediately replaced with fresh buffer pH 6.8. The experiment lasted 7 h. The sample from the donor cell was filtered (syringe nylon filter, 13 mm diameter, pore size 0.45 μm). The absence of BCL adsorption to the filter was approved by the comparison of the filtered and centrifuged samples. The concentrations of the samples were measured in a 96-well UV black plate (Costar) using a spectrophotometer (Spectramax 190; Molecular devices, Molecular Devices Corporation, San Jose, CA, USA).

As a result of the dissolution/permeation experiments, the kinetic curves of the dissolution/release, as well as the permeation profiles of pure BCL and BCL from the BCL/Ac-β-CD physical mixture and solid complex were obtained. Since the concentration of the donor solution is changing during the experiment (non-sink conditions), the flux (permeation rate), rather than the permeability coefficient (flux to initial concentration ratio) [38], was the parameter of interest used for the comparison of the behavior of the permeated samples.

#### 2.2.10. Apparent Solubility Determination

A sample containing an amount of 0.27 mg BCL or an equivalent amount of the BCL/Ac-β-CD physical mixture or solid complex was dispersed in 1.11 mL of pH 6.8 buffer medium (so that the CD concentration corresponded to the concentration in the V = 7 mL of the side-by-side cell) and mixed for 72 h in an air thermostat at 310.15 K. In addition, the experiments were carried out in the presence of PVP and HPMC in 0.01% (2.5 g·L^−1^) and 0.25% (0.1 g·L^−1^) concentrations, respectively. After the equilibrium was achieved, the suspensions were centrifuged at 310.15 K for 20 min at 12,000 rpm (Biofuge pico, Thermo Electron LED GmbH, Langenselbold, Germany). An aliquot of the saturated solution was taken at 310.15 K and then diluted (if necessary) by the pure buffer solution or the solution of Ac-β-CD, PVP, and HPMC. The absorbance was measured using a spectrophotometer (Cary 50 spectrophotometer Varian, Palo Alto, CA, USA, Software Version 3.00 (339)) with an accuracy of 2–4%. The experimental results are reported as an average value of at least three replicated experiments.

#### 2.2.11. In Vitro Permeation Experiments for the Apparent Permeability Coefficient Calculation

In order to determine the apparent permeability coefficients of the BCL and BCL/Ac-β-CD physical mixture and solid complex (from the solutions close to saturation), the following experimental setup was employed. The vertical-type Franz diffusion cell (PermeGear, Inc., Hellertown, PA, USA) of 7 mL volume of the donor compartment with an 0.785 cm effective surface area of the membrane was used. The regenerated cellulose membrane with a molecular weight cut-off (MWCO) of 12,000–14,000 Da (Standard Grade RC Dialysis Membrane, Flat Width 45 mm) was mounted between the donor and receptor chambers. The donor compartment (bottom) contained the solution, close to the saturation of pure BCL, BCL/Ac-β-CD solid complex, or the BCL/Ac-β-CD physical mixture for the comparison. The donor solution was mixed overnight and filtered, and the concentration was measured before the permeation experiment. The receptor chamber (top) was filled with 1 mL of pure buffer pH 6.8. The samples of 0.5 mL were withdrawn every 30 min and replaced with fresh pH 6.8 buffer and analyzed spectrophotometrically in a 96-well UV black plate (Costar) (Spectramax 190; Molecular devices, Molecular Devices Corporation, California, CA, USA). The experiments were carried out in sink conditions, meaning that the drug concentration in the acceptor chamber did not exceed 10% of the drug concentration in the donor chamber at any time. The steady state was set as even after the first hour of the experiments, and a steady state flux (*J*) of BCL across the membrane was derived from the linear part of the plot correlating to the amount of BCL permeated (*Q*) and time (*t*) taking into account the permeation area (*A*), whereas the permeability coefficient (*P_app_*) representing the steady state flux normalized by the initial concentration of the solution (*C*_0_) as:(1)$$P_{app}=J/C_{0}$$

#### 2.2.12. Calculation of the Complexation Thermodynamic Functions

For a complex with 1:1 stoichiometry, the complexation constant ($K_{C}$) was calculated by the following equation:(2)$$K_{C}=\frac{\left[D\cdot CD\right]}{\left[D\right]\left[CD\right]}$$
where [*D**⋅CD*] is the complex concentration, [*CD*] is the cyclodextrin concentration, and [*D*] is the drug concentration (M). From a *slope* of the linear phase solubility profile equal to $\left(K_{C}^{}\cdot S_{2}^{0}/(1+K_{C}^{}\cdot S_{2}^{0})\right)$ and intercept −$S_{2}^{0}$ the apparent stability constants (*Kc*) were estimated using the Higuchi–Connors equation [42]:(3)$$K_{C}^{}=\frac{slope}{S_{2}^{0}\cdot {\left(1-slope\right)}_{}},$$
where $S_{2}^{0}$ is the intrinsic solubility of the drug in the absence of CD. The complexation thermodynamic parameters, the standard change of the free Gibbs energy ($\Delta G_{C}^{0}$), enthalpy ($\Delta H_{C}^{0}$), and entropy ($\Delta S_{C}^{0}$) were derived as follows:(4)$$\Delta G_{C^{}}^{0}=-RT\ln K_{C}^{}$$
(5)$$\ln K_{C}^{}=-\frac{\Delta H_{C}^{0}}{RT}+\frac{\Delta S_{C}^{0}}{R}$$
where *T*—is the standard temperature equal to 298.15 K, and *R* is the universal gas constant.

## 3. Results

### 3.1. BCL Complex Formation with Me-β-CD and Ac-β-CD in Solution

The UV absorption spectra of the BCL and BCL/CD solutions were recorded, and they are depicted in Appendix A (Appendix A). As follows from the figure, the maximum BCL absorption spectrum is detected at λ = 270 nm, whereas in the presence of both CDs, the absorption maxima shifted to longer wavelengths (λ = 276 nm). The shift can be attributed to an essential modification of the BCL solvation shell from the interaction with CDs. No differences in the spectra of the methylated and acetylated CDs were observed. On the assumption of the UV spectrum’s sensitivity to the dielectric properties of the dissolution medium [45] and the lower dielectric constant of CD as compared to water, the shift of λ_max_ in the CD solutions is expected and approved for the formation of the complex. For the sake of comparison, the BCL absorption spectra in *n*-hexane (non-polar solvent) and ethanol, taken from our previous study [8], are also illustrated in Appendix A. Markedly, the shapes of all of the BCL absorption spectra were similar in all of the considered media. Evidently, the shifting of the BCL maximum, from water to the CDs solutions, towards the ethanol absorption peak testifies to the high probability of an interaction of the BCL molecule with the polar exterior of the CD molecule, which does not deny the affinity to the hydrophobic cavity.

The solubility of BCL was determined in the buffer solution at pH 6.8 and in the presence of the Me-β- and Ac-β-cyclodextrins of the 0.0025, 0.005, 0.01, and 0.015 M concentrations in the temperature range of 298.15–313.15 K. The PXRD patterns of parent BCL and solid residuals after the solubility experiments are illustrated in Appendix A. No phase transformations of BCL were detected. The phase solubility profiles are illustrated in Figure 2, and the solubility values are tabulated in the Appendix A (Appendix A).

The poor solubility of BCL in an aqueous solution was estimated elsewhere [4,5,8] earlier. Moreover, several attempts to improve the solubility have been undertaken (see the Introduction section). In the studies [4] with β-cyclodextrin and HP-β-cyclodextrin, approximately 2- and 2.7-fold solubility growth has been achieved by phase solubility experiments. In our investigation, a more efficient BCL solubility improvement by 9.7- and 20-fold using Me-β- and Ac-β- cyclodextrin, respectively, was estimated at 298.15 K. An elevated effect on the solubility improvement from the methylation of β-cyclodextrins (CH_3_-group in methylated CD) (log*K*^octanol/water^ ≈ −6), as compared to the parent and hydroxypropylated (log*K*^octanol/water^ = −14 and −11, respectively) ones, can most probably be attributed to their higher lipophilicity [46], which can facilitate the interactions of a highly lipophilic BCL molecule, not only with the hydrophobic CD cavity but also with the outer surface.

The linear dependences of the solubility were used for the stability constants ($K_{C}^{}$) calculations from the slope by Equation (3). The values of $K_{C}^{}$ are represented in Table 1.

First of all, the stability constant values estimated for both CDs belong to the range proposed by Szejtli of 200 to 5000 M^−1^ [47] for good potential bioavailability in oral dosage forms. A rather weak binding of BCL to Me-β-cyclodextrin (according to $K_{C}^{}$ in the range of 500–1000 M^−1^) and an approximately two-fold stronger binding to Ac-β-cyclodextrin was shown (Table 1). As follows, a smaller amount of Ac-β-CD is needed to solubilize BCL as compared to the methylated one. Taking into account that the amount of CD should be as little as possible in the case of a pharmaceutical application [48], Ac-β-CD seems to be advantageous as compared to Me-β-CD. For comparison, the stability constants of BCL with β-CD [4] and HP-β-CD [28] at 298.15 K were estimated as 318.87 M^−1^ and 161.36 M^−1^, respectively, which is a 6- and 12-fold lower than with Ac-β-CD, as estimated in the present study. As follows from Table 1, the complex formation processes are exothermic with both CDs, and the stability constants diminish with temperature growth. In order to reveal the driving forces of the complexation reaction, the thermodynamic parameters were calculated using Equations (4) and (5) and are listed in Table 2. The temperature dependences of the stability constants (Van’t Hoff plots) for the complexation of BCL with both CDs are illustrated in Appendix A.

The analysis of the thermodynamic complexation parameters has led to the following conclusions. The process is spontaneous and thermodynamically favorable, especially for the BCL/Ac-β-CD system. Interestingly, the ratio of the impacts from the enthalpy and entropy terms to the free Gibbs energy in the considered systems are significantly different. According to the absolute values of the $\Delta H_{C}^{0}$ and $T\Delta S_{C}^{0}$ terms, the complex formation is determined by the enthalpy in the case of acetylated CD. In its turn, the entropy-determined process in the system with methylated CD was revealed. Evidently, an almost five-fold greater negative enthalpy contribution testifies to a considerably stronger BCL interaction with Ac-β-CD as compared to Me-β-CD. Different signs of the entropy terms upon the complex formation indicate a rather different mode of hydrophobic interactions. For example, the complexation process between BCL and Me-β-CD (small negative $\Delta H_{C}^{0}$ and larger positive $\Delta S_{C}^{0}$) is accompanied by the reorganization of the solvation shells of the host and guest molecules and the dehydration of the CD cavity due to the release of the water molecules from the Me-β-CD cavity (mainly entropy driven complexation). According to Van der Jagt et al. [49], this case is attributed to a ‘nonclassical’ model of hydrophobic interaction, evidencing an essential impact of the hydrophobic effects during the complexation reaction. Both negative enthalpy and entropy contributions to the driving force of BCL/Ac-β-CD complex formation (maintaining an inequality |$\Delta H_{C}^{0}$| > |$\Delta S_{C}^{0}$| up to 3.8-fold) testify to the diminishing the role of the hydrophobic forces in favor of an essential impact of the van der Waals interaction as well as a decrease in structural freedoms during the complex formation (enthalpy driven process) [50]. As is evident from the obtained results based on the data in Table 1, Table 2, and Appendix A, Ac-β-CD was shown to be the better solubilizing agent for BCL than Me-β-CD.

### 3.2. Solid BCL/Ac-β-CD Complex

The next step of the study was devoted to the production of the solid BCL/Ac-β-CD complex. First of all, the stoichiometry was approved using Job’s continuous variation method [43] (see Appendix A). As follows from the figure, the stoichiometry of 1:1 was demonstrated by the symmetric dependence with an extremum at R = 0.5. Solid drug formulations are widely applicable in cases of oral administration. At the same time, for the enhancement of bioavailability, solubility/dissolution rate, stability, etc., the literature review evidences the advantage of the drug/CD complex rather than the simple physical mixture [51,52]. The solid complex BCL/Ac-β-CD (BCL/Ac-β-CD_gr) was prepared by a mechanical grinding procedure and characterized by DSC, PXRD, IR-spectroscopy, and SEM. For comparison, the ground BCL sample (BCL_gr) and BCL/Ac-β-CD physical mixture (BCL/Ac-β-CD_pm) were also prepared and characterized similarly to the complex. From the comparative analysis of all the samples, the physical state of BCL and the interactions with the CD molecule can be disclosed.

#### 3.2.1. Characterization of the Solid BCL/Ac-β-CD Complex

The DSC profile of BCL is shown in Figure 3, together with all of the prepared samples.

The similar broad endothermic peaks in the range of ~ 35÷100 °C are observed in the samples of pure Ac-β-CD, BCL/Ac-β-CD_pm, and BCL/Ac-β-CD_gr due to the CD dehydration. Notably, a more pronounced shift of the dehydration peak in Ac-β-CD (*T*_max_ = 54.2 ± 0.2 °C, *T*_onset_ = 29.4 ± 0.2 °C) to the higher temperatures in the ground complex (*T*_max_ = 57.4 ± 0.2 °C, *T*_onset_ = 28.7 ± 0.2 °C) as compared to physical mixture (*T*_max_ = 53.5 ± 0.2 °C, *T*_onset_ = 30.4 ± 0.2 °C) can be attributed to the stronger interaction of BCL with the CD cavity in the complex. The endothermic melting peak at *T*_max_ = 196.0 ± 0.2 °C (*T*_onset_ = 193.9 ± 0.2 °C) in raw BCL (form I as was stated in [8]) is very close to the ground sample (BCL_gr) (*T*_max_ = 194.6 ± 0.2 °C, *T*_onset_ = 192.9 ± 0.2 °C) and maintaining the crystallinity. In the physical mixture with Ac-β-CD, the peak maximum (*T*_max_ = 191.9 °C ± 0.2 °C) was broadened: *T*_onset_ = 185.7 ± 0.2 °C instead of 193.9 ± 0.2 °C for the pure BCL sample. Importantly, only a very small peak of BCL (*T*_max_ = 194.7 ± 0.2 °C, *T*_onset_ = 194.0 ± 0.2 °C) remains after the 1 h grinding procedure was applied to the BCL/Ac-β-CD physical mixture (∆*H_fus_* = 9.9 ± 0.2 J·g^−1^) to produce the BCL/Ac-β-CD solid complex (∆*H_fus_* = 0.0182 ± 0.5 J·g^−1^). In order to evaluate the crystallinity of different samples, the equation borrowed from the literature [53] was applied:(6)$$\%crystallinity=\left(\frac{\Delta H_{fus}(sample)}{\Delta H_{fus}(BCLraw)\cdot W}\right)\cdot 100$$
where $\Delta H_{fus}(sample)$ and $\Delta H_{fus}(BCLraw)$ are the fusion enthalpy of the treated sample and raw BCL, respectively, W is the weight fraction of BCL in the treated sample. The effect of processing on the percent of crystallinity of the investigated samples is demonstrated in Figure 4.

The fusion enthalpy of the raw BCL calculated from the DSC experiment was assumed to be 100% crystallinity. As follows from Figure 4, the crystallinity of raw BCL was reduced by only 2.5% after the 1 h grinding. At the same time, a 1.8- and 952.4-fold crystallinity reduction was demonstrated for the BCL/Ac-β-CD_pm and BCL/Ac-β-CD_gr preparations, respectively. Evidently, the treatment of BCL in a planetary mill maintains the crystalline nature of the compound. At the same time, the simple mixing of BCL with Ac-β-CD leads to a diminished peak sharpness accompanied by a reduction in the enthalpy and percent of crystallinity. A dramatic decrease in the % crystallinity of the ground BCL/Ac-β-CD sample obviously comes from the formation of the amorphous inclusion complex between BCL and Ac-β-CD (through the molecular encapsulation of the drug inside the CD cavity) rather than from the effect of mechanical grinding. Moreover, this result, to a large extent, comes from the amorphous nature of Ac-β-CD and is indicative of more effective BCL interactions in the solid state. Rushing ahead, we can predict an advantageous solubility/dissolution behavior of the BCL/Ac-β-CD_gr complex (in Section 3.2.3), undoubtedly, due to the maximal amorphization of the ground sample. As was emphasized in [51], the enhancement of the oral bioavailability of the complex (as compared to drug alone and physical mixture with CD) is likely related to the increased enhancement of the dissolution kinetics observed when the complex, rather than the physical mixture, is utilized.

The diffraction patterns of all of the investigated samples are illustrated in Figure 5. The PRXD pattern of Ac-β-CD, representing a diffuse halo, is characteristic of the amorphous structure. The crystalline state of raw BCL (black line in Figure 5) remains practically the same in the ground sample (black dash-dot line), with the sharpest peaks at 2*θ* angles of 12.1°, 16.8°, and 23.6°. The physical mixture of BCL and CD essentially demonstrates reduced crystallinity. The crystallinity of the ground sample was decreased to a greater extent as compared to the physical mixture: the only peak at 12.1° (2*θ*) remains the most pronounced in the ground sample of the BCL/Ac-β-CD complex; the peaks at 16.8° and 23.6° are only scarcely visible. This result approves the DSC analysis, showing a slight amount of crystalline BCL in the complex with CD.

In search of the interactions between BCL and Ac-β-CD, the FTIR spectra were recorded (Figure 6). As is evident from the figure, the main peaks of the raw BCL detected in our work are in agreement with those reported in the literature [4,30]. Ac-β-CD demonstrates the characteristic peaks at 3412 cm^−1^ (O-H), 2925 cm^−1^ (C-H), 1748 cm^−1^ (acetyl group vibration assigned to carbonyl C = O stretching of ester), 1640 cm^−1^ (H-O-H bending), 1375 cm^−1^ (-C-H bending vibration), 1240 cm^−1^ (C-O stretching of the acetyl group), 1155 cm^−1^ (C-O), and 1037 cm^−1^ (C-O-C). As was shown by Liu et al. [54], the three ester bonds at 1748, 1375 (intensified by the acetylation), and 1240 cm^−1^ come from the acetylation of parent β-CD. The comparative analysis of the investigated samples revealed some similarities between the BCL/Ac-β-CD physical mixture and the ground complex; no new peaks were detected in the ground product, which can be attributed to the absence of the covalent interactions between the compound and CD, as was shown by Ford [55]. The peaks at 3577 (O-H), 3057 (C-H aromatic), 2939 (C-H aliphatic asymmetric), and 1689 (C=O) cm^−1^ of BCL completely disappeared in the spectra of both the physical mixture and the complex. In their turn, the bands at 2231 cm^−1^ (C≡N), 1595 cm^−1^ (C=C aromatic), and 1323 cm^−1^ (S=O) are visible in both the mixed and ground systems, and the peak at 1137 cm^−1^ (C-F of CF_3_) shifted to 1141 cm^−1^ in both pm and the complex. Meanwhile, some distinguishable features between the physical mixture and the complex should be marked. The peak of BCL at 3336 cm^−1^ (N-H) is fully lacking in the spectrum of the ground complex, showing some impact of the hydrogen bonding to the complex formation. The band at 844 cm^−1^ (*p* substituted benzene), maintained in the physical mixture, shifted to 846 cm^−1^ in the complex. A different mode of BCL’s interaction with Ac-β-CD in the ground, solid complex as compared to the kneaded products of BCL/β-CD [4] and BCL/HP-β-CD [30] was revealed as follows from the FTIR spectra characteristics. This fact clearly demonstrated the importance of both the preparation technique and the type of CD used.

The morphology of the samples: BCL_raw, BCL_gr, BCL/Ac-β-CD_pm, and BCL/Ac-β-CD_gr were examined by SEM analysis. The results are illustrated in Figure 7. The three-dimensional BCL crystals with a raw surface and different sizes (Figure 7(1a–1d)) developed essentially smaller (Figure 7(2a–2d)) dimensions after the mechanical grinding. The irregularly shaped structure of Ac-β-CD is demonstrated in Figure 7(5a–5d). In the physical mixture (Figure 7(3a–3d)), both the specific particles of BCL and CD are visible. At the same time, the morphology of the particles in the co-ground system (Figure 7(4a–4d)) is different from both of the original components, which allows for the proposal of the formation of the complex between BCL and Ac-β-CD. The average size of the complex is approximately 10 micrometers.

#### 3.2.2. Equillibrium Solubility of the BCL/Ac-β-CD as Solid Samples

The analysis of the results on the equilibrium solubility of the BCL solid samples in pH 6.8 buffer and in the presence of PVP and HPMC at 310.15 K led to the following solubility order: BCL_raw = BCL_gr < BCL/Ac-β-CD_pm < BCL/Ac-β-CD_gr ≤ BCL/Ac-β-CD_gr (PVP) < BCL/Ac-β-CD_gr (HPMC) (Figure 8).

First of all, the unchanged crystallinity of the ground BCL was shown and resulted in the same solubility as BCL_raw. The solubilities of the BCL physical mixture and the ground complex are 1.15- and 1.38-fold, respectively, higher than the unprocessed BCL. The presence of PVP in the buffer solution practically does not affect the BCL/Ac-β-CD_gr solubility; at the same time, in the HPMC solution, a 1.88-fold solubility increase was observed. Evidently, the nature of the polymer is crucial for the solubility enhancement in a specific system (a complex character of the interaction between all the components plays a significant role). For example, it was emphasized [56,57] that the water-soluble cellulose derivatives can form complexes with cyclodextrin, thus changing the physicochemical properties of the drug.

#### 3.2.3. Non-Sink Dissolution/Permeation Behavior of the BCL Solid Samples

It was stated [58] that cyclodextrins increase bioavailability when dissolution, rather than permeation through the intestinal membrane, is the rate-limiting step in drug absorption. It should be noted that an examination of the kinetic dissolution profiles of the pure or treated drug substance, and especially drug solubility-enabling formulations under conditions where pH, stirring rate, and temperature are kept constant, may mimic dynamic in vivo dissolution processes better than equilibrium solubility studies. Moreover, in cases of a significant increase in the solubility of strongly-lipophilic compounds using the solubility enhancing approaches (ex, amorphous and solid dispersions), too much but short supersaturation may occur in the gastrointestinal lumen [59], which should be controlled. On the one hand, the increased intraluminal drug concentration can result in an enhanced flux across the intestinal wall [59], but on the other one, the state of the drug molecule in the solubility-enabling formulations may contribute to bioavailability alleviation [60]. In view of this, the simultaneous assessment of dissolution and permeation is very useful. We carried out the dissolution/permeation experiments with the help of a side-by-side cell. The regenerated cellulose membrane with a molecular weight cut-off (MWCO) of 12,000–14,000 Da was shown to be useful for the design of drug–cyclodextrin formulations [34]. As opposed to lipophilic biological membranes, not only the drug but also the drug-CD complexes can diffuse through the membrane [44]. Since the transport through the cellulose membrane is determined by the diffusion of the free drug molecules plus the CD complexes, the precipitation of poorly soluble compounds in the acceptor chamber is prevented. The dissolution/permeation (D/P) experiments were carried out in a 7 mL/7 mL volume side-by-side cell in order to simultaneously monitor the dissolution rate and permeation flux under non-sink conditions as a function of time. The dissolution profiles of the raw BCL, physical mixture, and the complex with Ac-β-CD in pH 6.8 buffer at 310.15 K are illustrated in Figure 9a. Additionally, the profiles of the complex in the presence of 0.01% PVP and 0.25% HPMC in buffer solution are presented.

The dissolution of raw BCL is a very slow process; a somewhat higher release from the BCL/Ac-β-CD_pm sample was measured. As is evident from Figure 9a, the amorphous solid dispersion of the BCL/Ac-β-CD complex demonstrated the essentially faster dissolution (as compared to the free drug and physical mixture), possibly due to the different properties of the complex, for example, enhanced hydrophilicity and a greater extent of amorphization, allowing water molecules to break down the interactions between the drug molecules easily, thus, increasing the driving force of the dissolution. Moreover, the authors [51] underlined that the formation of the complex results in an increased surface area of the complex as compared to the physical mixture and drug alone. As is evident from Figure 9a, the release of BCL from the complex is followed by the supersaturation that is often characteristic of amorphous systems [61]. On the one hand, supersaturation usually provides advantageous kinetic behavior in vivo, but being a metastable state, it should be stabilized in order to achieve adequate absorption and bioavailability and delay the crystallization (precipitation) of the drug [59,62]. To this end, a whole number of precipitation inhibitors are used. Among them, polyvinilpyrrolidone (PVP) and cellulose derivatives are of paramount importance [40]. In order to apply this approach to the BCL/Ac-β-CD_gr complex, we use PVP in a concentration of 0.01%, as proposed by Lindfors et al. [63] for the best inhibition of BCL crystallization, and hydroxypropyl methylcellulose (HPMC) in the concentration of 0.25% as a somewhat average from several studies with different drugs and HPMC as a precipitation inhibitor [61,64,65].

For comparison, the characteristics of the dissolution behavior of the samples were determined and listed in Table 3: the time at which the maximal BCL concentration is achieved (time_Cmax_), the BCL amount dissolved at 20, 60, and 120 min (*C*_20_, *C*_60_, and *C*_120_), the area under the dissolution curve of the tested sample from zero time to the last time point of the dissolution experiment (t) ($AUC_{C(t)}$), t = 420 min, and the excipient gain factor describing the extent to which excipients stabilize supersaturation (EGF). The EGF parameter was calculated as follows [66]:(7)$$EFG=\frac{AUC_{C(t)}^{exipient}}{AUC_{C(t)}^{WITHOUTexcipient}}$$
where $AUC_{C(t)}^{excipient}$ and $AUC_{C(t)}^{WITHOUTexcipient}$—is the area under the dissolution curve of a sample in the presence and in the absence of the excipient, respectively.

As Figure 9a and Table 3 show, the BCL/Ac-β-CD_gr complex has the fastest dissolution rate in the presence of C = 2.5 g·L^−1^ HPMC. Namely, 27.7 μg·mL^−1^ was dissolved within 60 min, which is 21.3-, 6.6-, 2.9-, and 1.1-fold greater than for BCL_raw, BCL_pm, BCL/Ac-β-CD_gr, and BCL/Ac-β-CD_gr in the presence of PVP 0.1 g·L^−1^, respectively. Markedly, the maximal BCL concentration for BCL/Ac-β-CD_gr in pure buffer and in the PVP solution at the times of 20 and 30 min, respectively, was detected. This time for the BCL/Ac-β-CD_gr sample in the HPMC solution is 90 min. As was shown by Lindfors et al. [63], the effect of PVP on the prolongation of the supersaturated state is based on the adsorption of the polymer to the crystals and the formation of a physical barrier between the solution and the drug crystal lattice. In its turn, the role of the polymeric cellulose derivatives as precipitation inhibitors was demonstrated in many studies, including those dealing with non-sink conditions [67,68], in which the delay of the precipitation was attributed to the formation of the drug-polymer associates due to the surface interactions between them, as well as the hydrogen binding between the donor and acceptor atoms of both the drug and polymer [64]. In our experiments, the effectiveness of the BCL dissolution (release) processes (illustrated by the $AUC_{C(t)}$ parameter) were estimated to be 7.8-, 5.8-, 3.0-, and 1.8-fold higher for BCL/Ac-β-CD_gr (HPMC), BCL/Ac-β-CD_gr (PVP), BCL/Ac-β-CD_gr, and BCL/Ac-β-CD_pm, respectively, as compared to BCL_raw sample. The excipient gain factor (EGF), calculated for the dissolution of the BCL/Ac-β-CD_gr sample, was shown to be 2.59 for BCL/Ac-β-CD_gr in the presence of HPMC, which is 1.3-fold greater as compared to PVP. From the experimental dissolution results, it can be concluded that the formation of BCL ground complex with Ac-β-CD enhances the dissolution rate of the compound. The presence of HPMC as a stabilizer of the supersaturation state is promising and seems to be a useful tool for manufacturing BCL pharmaceutical formulations.

As was underlined above, the permeation properties are no less important than the dissolution for the bioavailability evaluation. Moreover, bearing in mind that the presence of the excipients, both in the content of the solid dispersions [69] and in solution, can modify the permeation, often unpredictably, we tried to consider the solubility/dissolution processes simultaneously in a combined setup. The permeation profiles of the investigated formulations are illustrated in Figure 9b. The concentration of BCL on the acceptor side, which permeated through the membrane, is significantly higher for the ground complex (especially if PVP or HPMC are in the solution) as compared to the physical mixture and the crystalline BCL. The flux during the first 30 min was calculated to be 1.59 × 10^−6^, 1.79 × 10^−6^, 4.52 × 10^−6^, 8.15 × 10^−6^, and 6.11 × 10^−6^ μM·cm^−2^·s^−1^ for BCL_raw, BCL/Ac-β-CD_pm, BCL/Ac-β-CD_gr, BCL/Ac-β-CD_gr (PVP), and BCL/Ac-β-CD_gr (HPMC), respectively. Markedly, the maximal value for the BCL/Ac-β-CD_gr (PVP) sample was in agreement with the fastest dissolution rate (Figure 9a) during this period. At the same time, after 120 min of the experiment, the permeation curve of BCL/Ac-β-CD_gr (HPMC) was higher than the BCL/Ac-β-CD_gr (PVP) curve, slightly increasing the steady state flux (5.67 × 10^−7^ and 6.11 × 10^−7^ μM·cm^−2^·s^−1^, respectively). Interestingly, the maximal steady-state flux was derived for BCL/Ac-β-CD_gr sample (8.99 × 10^−7^ μM·cm^−2^·s^−1^), most probably due to the practically unchanged donor concentration during this period of time, whereas in the presence of both polymers the donor concentration constantly decreases (Figure 9a) as a result of the prolonged supersaturation.

#### 3.2.4. Permeability Experiments in Sink Conditions, Determination of the Steady State BCL Permeability Coefficients

The permeability coefficient is an important numerical parameter that reveals the impact of the composition of the drug formulations, including different excipients. To this end, the cellulose membrane, often used for this purpose [44], was applied. The permeability coefficients of the BCL solid samples were determined in a vertical type 7 mL Franz diffusion cell with a constant concentration of BCL in the donor solution (as opposed to Section 3.2.3). The experimental data (the values of the donor concentration, steady-state flux, and permeability coefficients) are tabulated (Appendix A). The values of the permeability coefficients vs. the steady-state fluxes are also illustrated in a diagram (Figure 10).

The lowest steady-state flux and the highest permeability coefficient for raw BCL (Figure 10a) are due to the lowest concentration of the compound in the donor solution but the larger portion of the free (unbound with CD) BCL molecules more readily passed through the membrane. Rather close values of the fluxes through the membrane of all the studied samples (BCL/Ac-β-CD_pm > BCL/Ac-β-CD_gr > BCL_raw) but a 1.3-fold lower permeability coefficient of the ground sample as compared to the physical mixture is possibly determined by the more pronounced interactions of BCL with CD in the complex, resulting in the lower diffusivity and thus inhibiting the movement across the barrier. The comparison of the permeability from the solution with the results of the combined dissolution/permeation experiments (both at steady state) (Figure 10b) showed that in the last case, the steady-state fluxes are essentially lower for the raw BCL and physical mixture with CD. As opposed to the similar (within the experimental error) flux values derived from the linear part of the permeability vs. time dependence from the ground sample (BCL/Ac-β-CD_gr) were obtained, most probably due to the positive impact of the supersaturation on the permeation.

In the next step, we compare the BCL permeability coefficient measured in the present study in the Franz cell through the cellulose membrane with the one evaluated across the PermeaPad barrier (biomimetic lipophilic barrier based on the liposome layer on the support sheet) in our earlier work [8]. *P*_app_ (cellulose membrane) was shown to be 3.6-fold higher than *P*_app_ (PermeaPad), obviously due to the significant resistance of the lipophilic layer of the barrier.

## 4. Conclusions

A bicalutamide/acetyl-β-cyclodextrin solid complex of 1:1 stoichiometry ($K_{C}^{}$ = 1919.3 ± 61.8 M^−1^ at 298.15 K) was prepared for the first time with the of improving its dissolution rate and permeation flux through the membrane. The average size of the complex was estimated by the SEM analysis at approximately 10 micrometers. In the non-sink dissolution/permeation test using the regenerated cellulose membrane, the complex demonstrated an essentially faster dissolution (accompanied by the supersaturation) as compared to the free drug and physical mixture, namely, $AUC_{C(t)}$(BCL/Ac-β-CD_gr) > $AUC_{C(t)}$ (BCL/Ac-β-CD_pm) > $AUC_{C(t)}$(BCL_raw)—a 1.6- and a 3.0-fold, respectively. The effectiveness of the BCL/CD complex release was estimated to be 7.8- and 5.8-fold higher in the presence of HPMC and PVP (the precipitation inhibitors), respectively, as compared to pure buffer. The excipient gain factor was shown to be 2.59 in HPMC, which is 1.3-fold greater than in the PVP solution. The elevated fluxes of the BCL/Ac-β-CD complex through the membrane from the polymers solutions were demonstrated to be in line with the dissolution profiles, most probably due to the variations of the donor solution concentration. The obtained results demonstrated the advantage of the dissolution–permeation experiments and disclosed that the presence of HPMC as a stabilizer of the supersaturation state was shown to be promising and useful for the optimization of BCL pharmaceutical formulations manufacturing.

## Figures and Tables

**Figure 1 pharmaceutics-14-01472-f001:**
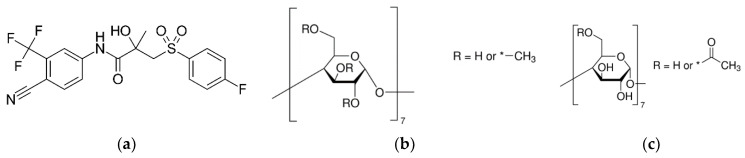
Structures of bicalutamide (BCL) (**a**), methylated β-CD (Me-β-CD) (**b**), acetylated β-CD (Ac-β-CD) (**c**); *—means the substituent in the cyclodextrin molecule.

**Figure 2 pharmaceutics-14-01472-f002:**
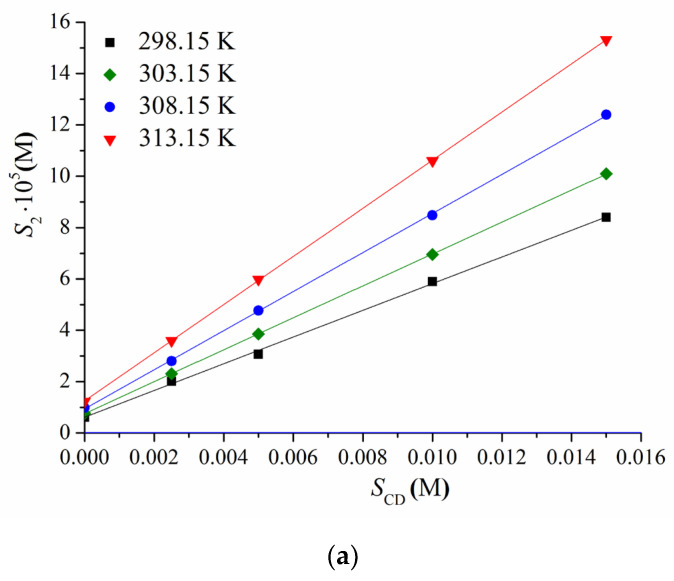
Phase solubility diagrams of BCL with Me-β- (**a**) and Ac-β- (**b**) cyclodextrins at different temperatures: 298.15 K—black, 303.15 K—green, 308.15 K—blue, and 313.15 K—red color.

**Figure 3 pharmaceutics-14-01472-f003:**
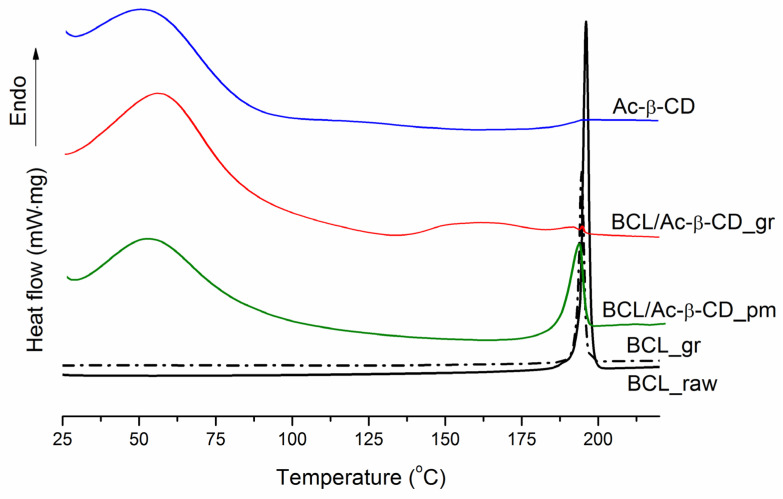
DSC profiles of BCL raw material (BCL_raw black solid line), BCL ground sample (BCL_gr black dash-dot line), BCL physical mixture with Ac-β-CD (BCL/Ac-β-CD _pm green solid line), BCL with Ac-β-CD ground sample (BCL/Ac-β-CD_gr red solid line), Ac-β-CD raw material (Ac-β-CD blue solid line).

**Figure 4 pharmaceutics-14-01472-f004:**
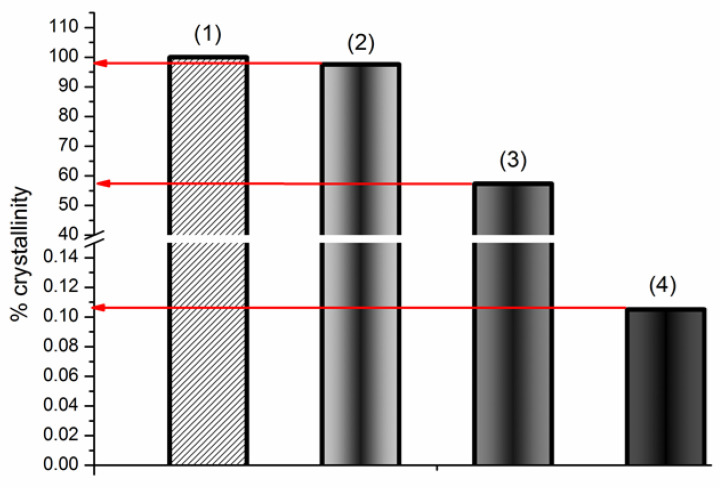
Effect of grinding (1 h) on raw BCL crystallinity: BCL raw (1), BCL_gr (2), BCL/Ac-β-CD_pm (3), BCL/Ac-β-CD_gr (4). The % crystallinity is calculated by Equation (6).

**Figure 5 pharmaceutics-14-01472-f005:**
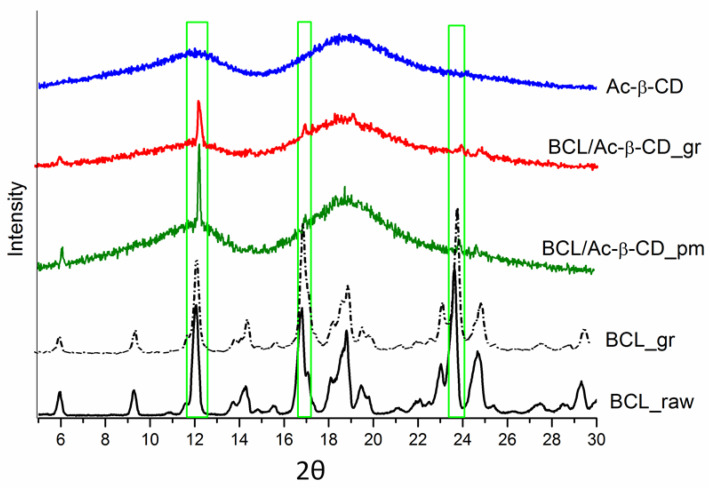
PXRD patterns of BCL raw material (BCL_raw black solid line), BCL ground sample (BCL_gr black dash-dot line), BCL physical mixture with Ac-β-CD (BCL/Ac-β-CD_pm green solid line), BCL with Ac-β-CD ground sample (BCL/Ac-β-CD_gr red solid line), Ac-β-CD raw material (Ac-β-CD blue solid line).

**Figure 6 pharmaceutics-14-01472-f006:**
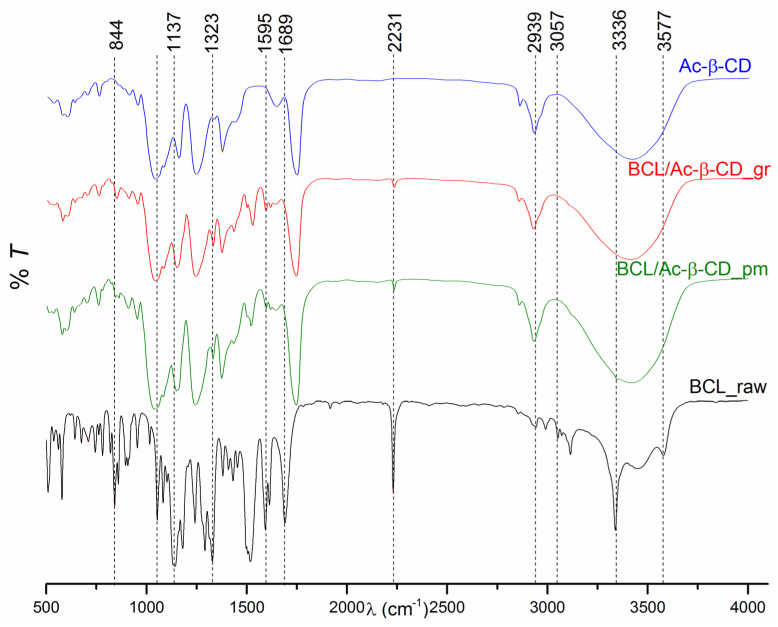
FTIR spectra of BCL raw material (BCL_raw black line), BCL physical mixture with Ac-β-CD (BCL/Ac-β-CD_pm green line), BCL with Ac-β-CD ground sample (BCL/Ac-β-CD_gr red line), Ac-β-CD raw material (Ac-β-CD blue solid line).

**Figure 7 pharmaceutics-14-01472-f007:**
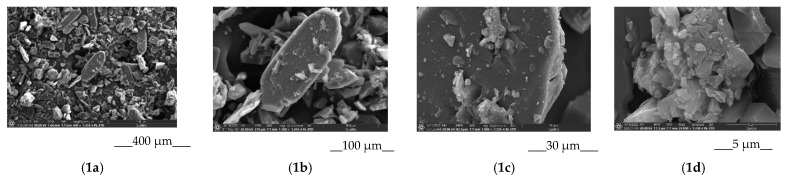
SEM images of raw BCL (**1a,1b,1c,1d**), ground BCL (**2a,2b,2c,2d**), BCL/Ac-β-CD_pm (**3a,3b,3c,3d**), BCL/Ac-β-CD_gr (**4a,4b,4c,4d**), Ac-β-CD (**5a,5b,5c,5d**). Indexes (**a**), (**b**), (**c**), and (**d**) are referred to (×400), (×1300), (×5000) and (×24,000), respectively.

**Figure 8 pharmaceutics-14-01472-f008:**
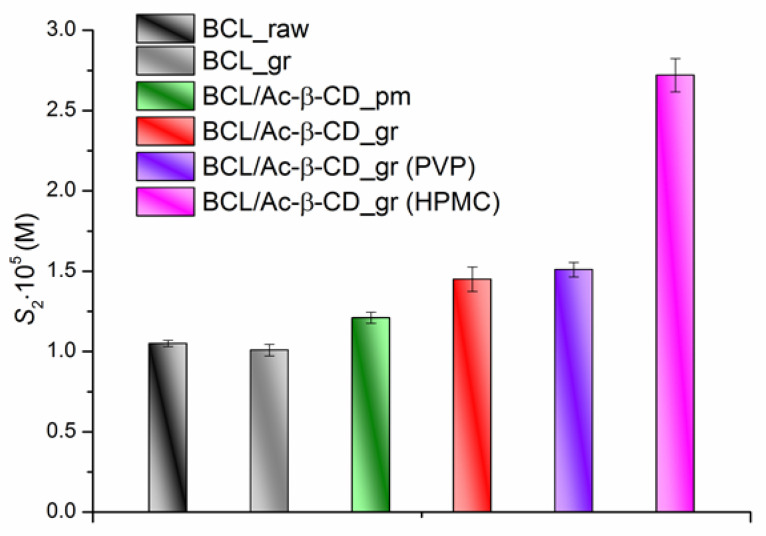
The equilibrium solubility of the BCL solid samples in pH 6.8 buffer at 310.15 K. BCL_raw (1.05 × 10^−5^ M) = BCL_gr (1.01 × 10^−5^ M) < BCL/Ac-β-CD_pm (1.21 × 10^−5^ M) < BCL/Ac-β-CD_gr (1.45 × 10^−5^ M) ≤ BCL/Ac-β-CD_gr (PVP) (1.51 × 10^−5^ M) < BCL/Ac-β-CD_gr (HPMC) (2.72 × 10^−5^ M).

**Figure 9 pharmaceutics-14-01472-f009:**
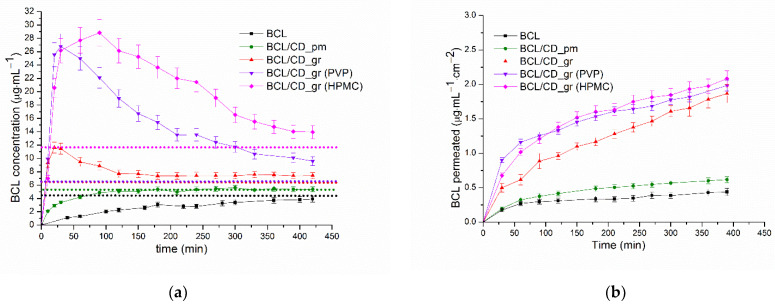
Dissolution (**a**) and permeation (**b**) profiles of the studied BCL samples using the combined dissolution/permeation setup at 310 K. The BCL amounts related to the equilibrium solubility in the respective conditions are indicated by the dotted lines.

**Figure 10 pharmaceutics-14-01472-f010:**
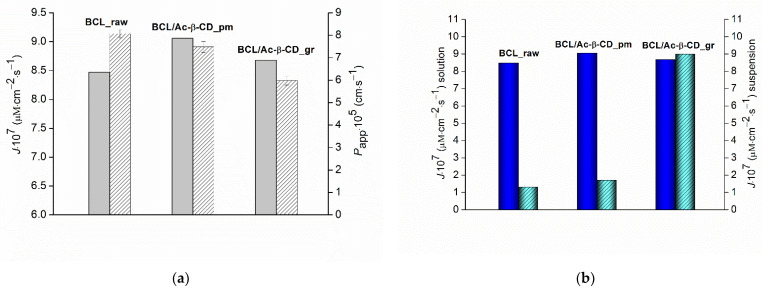
(**a**) The steady-state fluxes in sink conditions (left grey columns) and permeability coefficients (right dashed columns) from the experiments in the Franz diffusion cell; (**b**) the steady-state fluxes in non-sink conditions from the solution (Franz diffusion cell) (left blue columns) and from the suspension (side-by-side cell) (right cyan columns) of BCL raw material (BCL_raw), BCL physical mixture (BCL/Ac-β-CD_pm) and BCL ground sample (BCL/Ac-β-CD_gr) (buffer pH 6.8, 310.15 K).

**Table 1 pharmaceutics-14-01472-t001:** Apparent stability constant ($K_{C}^{}$) of BCL complexes with Me-β- and Ac-β-cyclodextrin at different temperatures and pH 6.8.

T (K)	$K_{C}^{}(M^{-1})$	
	Me-β-Cyclodextrin	Ac-β-Cyclodextrin
298.15	849.2 ± 40.3	1919.3 ± 61.8
303.15	812.8 ± 37.5	1613.3 ± 35.3
308.15	782.5 ± 34.2	1332.5 ± 43.2
315.15	769.8 ± 33.6	1187.0 ± 28.7

**Table 2 pharmaceutics-14-01472-t002:** Complexation constant ($K_{C}^{}$) at 298.15 K and the standard (at 298.15 K) thermodynamic parameters of complex formation reaction for BCL/Me-β-CD and BCL/Ac-β-CD: change of the standard Gibbs free energy ($\Delta G_{C}^{0}$), enthalpy ($\Delta H_{C}^{0}$), and entropy ($\Delta S_{C}^{0}$) in buffer solution (pH 6.8) at 298.15 K.

System	*K_C_* (M^−1^)	$\Delta G_{C}^{0}\;(\mathrm{kJ}\cdot {\mathrm{mol}}^{-1})$	$\Delta H_{C}^{0}\;(\mathrm{kJ}\cdot {\mathrm{mol}}^{-1})$	$298.15\cdot \Delta S_{C}^{0}\;(J\cdot K^{-1}\;{\mathrm{mol}}^{-1})$
BCL/Me-β-CD ^a^	849.2 ± 40.3	−16.7 ± 0.3	−5.2 ± 0.6	11.5 ± 1.6
BCL/Ac-β-CD ^b^	1919.3 ± 61.8	−18.7 ± 0.4	−25.4 ± 1.6	−6.7 ± 0.6

^a^ $\ln K_{C}^{}=4.7(\pm 0.3)+622(\pm 77)1/T;R=0.985$; ^b^ $\ln K_{C}^{}=-2.7(\pm 0.6)+3052(\pm 187)1/T;R=0.996$.

**Table 3 pharmaceutics-14-01472-t003:** Characteristics of the dissolution behavior: time_Cmax_, *C*_20_, *C*_60_, *C*_120_, and DPP for the investigated solid samples of BCL.

System	^1^ time_Cmax_ (min)	^2^*C*_20_ (μg·mL^−1^)	^2^*C*_60_ (μg·mL^−1^)	^2^*C*_120_ (μg·mL^−1^)	^3^$AUC_{C(t)}$ (μg·min)	^4^ EGF
BCL_raw		0.56	1.30	2.27	1113.99	
BCL/Ac-β-CD_pm		2.92	4.20	5.07	2058.18	
BCL/Ac-β-CD_gr	20	11.63	9.46	7.70	3346.34	
BCL/Ac-β-CD_gr (PVP)	30	25.54	24.98	19.00	6451.43	1.93
BCL/Ac-β-CD_gr (HPMC)	90	20.57	27.70	26.13	8653.46	2.59

^1^ time_Cmax_—the time at which the maximal BCL concentration is achieved; ^2^
*C*_20_, *C*_60_, and *C*_120_—the BCL amount dissolved at 20, 60, and 120 min; ^3^
$AUC_{C(t)}$—the area under the dissolution curve of the tested sample from zero time to the last time point of the dissolution experiment (t); ^4^ EGF—excipient gain factor.

## Data Availability

The results obtained for all experiments performed are shown in the manuscript and Appendix A, the raw data will be provided upon request.

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
