# Peer review of "Another Move towards Bicalutamide Dissolution and Permeability Improvement with Acetylated β-Cyclodextrin Solid Dispersion"

_pharmaceutics, 2022, doi:10.3390/pharmaceutics14071472_

Round 1

Reviewer 1 Report

In the present manuscript the authors report the synthesis and characterization of a complex between modified CDs and bicalutamide. The work is interesting since it provides some interesting scientific findings about new pharmaceutical formulations. It is well performed and in my opinion it deserves to be published on Pharmaceutics after appropriate revision.

1. In the introduction part the authors stated that the interaction between CD and biculatamide is already reported. To the light of this, the authors should better emphasize the novelty of their work in comparison to others already published.

2. Some further considerations about the crystallinity of the drug after CD complexation should be made. In general, when a drug is complexed its crystallinity decreases as a consequence of the interaction. I believe that it is the case reported in the present manuscript.

3. SEM images with higher magnification should be provided. Furthermore please highlight the scale bar of the different images for clarity.

Author Response

Reply to Reviewer_1

In the present manuscript the authors report the synthesis and characterization of a complex between modified CDs and bicalutamide. The work is interesting since it provides some interesting scientific findings about new pharmaceutical formulations. It is well performed and in my opinion it deserves to be published on Pharmaceutics after appropriate revision.

Comment:

  1. In the introduction part the authors stated that the interaction between CD and biculatamide is already reported. To the light of this, the authors should better emphasize the novelty of their work in comparison to others already published.

Reply:

The novelty of the presented investigation has been additionally underlined in the Introduction (Line 94-96, 107-108).

Comment:

  1. Some further considerations about the crystallinity of the drug after CD complexation should be made. In general, when a drug is complexed its crystallinity decreases as a consequence of the interaction. I believe that it is the case reported in the present manuscript.

Reply:

Additional considerations on the crystallinity reduction after BCL complexation with Ac-b-CD have been introduced in the discussion of the crystallinity issue (Line 419-425).

Comment:

  1. SEM images with higher magnification should be provided. Furthermore please highlight the scale bar of the different images for clarity.

Reply:

SEM images with higher magnification (×24 000) have been provided (Figure 7 (1-5d)). The scale bars have been added under the images for clarity.

Reviewer 2 Report

1.       Line 17-18: Abstract: Mention the grades of HPMC and PVP or write molecular weight in bracket

2.       In introduction of bicalutamide include: Dosage form available and Dose of drug to be delivered in one dose (Imp for under standing need of enhancing solubility), also mentioned solubility reported in official books if any.

3.       Add, reported solubility at different pH (Important to understand gastric behaviour)

4.       Add, Literature on solubility enhancement work don on this drug and compare how this work is different or novel

5.       Conclusion need to be rewrite: In present condition it is more like a summary. Point out overall conclusion in few sentences.  

6.       Cited work is bit old, focus on last five or ten years work for better corelation. (Specially introductory part)

Author Response

Reply to Reviewer_2

Comment:

Line 17-18: Abstract: Mention the grades of HPMC and PVP or write molecular weight in bracket

Reply:

The molecular weights of HPMC and PVP have been indicated in the Abstract (Line 17-18).

Comment:

In introduction of bicalutamide include: Dosage form available and Dose of drug to be delivered in one dose (Imp for under standing need of enhancing solubility), also mentioned solubility reported in official books if any.

Reply:

The information on the dose dosage form, dose and data on the BCL solubility available has been added to Introduction (Line 48-51).

Comment:

Add, reported solubility at different pH (Important to understand gastric behaviour)

Reply:

The literature survey contains mainly the information on the solubility in unbuffered water and buffer pH 7.4. At the same time, the pKa value of BCL at approximately 11.5 implies the solubility in aqueous media to be independent on pH in the biologically relevant region. The respective text has been inserted in the Introduction(Line 53-55).

Comment:

Add, Literature on solubility enhancement work don on this drug and compare how this work is different or novel

Reply:

Based on the existed literature sources on the BCL solubility enhancement the difference and novelty of the presented investigation has been underlined in the Introduction (Line 94-96, 107-108).

Comment:

Conclusion need to be rewrite: In present condition it is more like a summary. Point out overall conclusion in few sentences. 

Reply:

The conclusion has been rewritten according the comment (Line 612-627).

Comment:

Cited work is bit old, focus on last five or ten years work for better corelation. (Specially introductory part)

Reply:

The literature cited has been focused on the studies of last five or ten years, several references have been added to Introduction in the Introduction (Line 668-670, 687-690, 693-695 in the reference list).

Reviewer 3 Report

The manuscript is presented well. From the literature it is understood that the authors have been working on bicalutamide for many years. I have few comments below.

2.2.1. UV-spectroscopy: Please mention the calibration range

2.2.6. PXRD analysis: Did authors use the same quantity for all samples? Did you consider the particle size during the analysis? Please mention the quantity of the sample used for the analysis. 

Representing the diffraction patters are very important for this study, please provide PXRD patterns. Also provide overlaid IR spectra

2.2.9. Dissolution/permeation setup in a side-by-side cell; Why did the authors choose pH 6.8? I understand it is intestinal pH, is there any concept behind this?

2.2.11. Again, what is the concept behind  pH 6.8 in receiver compartment, why not pH 7.4 or close to blood pH

Why did authors use both sink and non sink conditions for permeation studies?

Did the authors store the samples at 40C/75RH to notice the changes in morphology and solubility?

Author Response

Reply to Reviewer_3

The manuscript is presented well. From the literature it is understood that the authors have been working on bicalutamide for many years. I have few comments below.

Comment:

2.2.1. UV-spectroscopy: Please mention the calibration range

Reply:

The calibration range of 6·10-6 ÷ 2·10-6 M has been added to the solubility determination using the UV-spectroscopy (Line 178).

Comment:

2.2.6. PXRD analysis: Did authors use the same quantity for all samples? Did you consider the particle size during the analysis? Please mention the quantity of the sample used for the analysis.

Reply:

The quantity of the sample used for the PXRD analysis was 50 mg (the respective information has been inserted in Methods PXRD analysis) (Line 209-210). The particle size was not considered during the PXRD analysis, only in morphological study with the SEM experiments.

Comment:

Representing the diffraction patters are very important for this study, please provide PXRD patterns. Also provide overlaid IR spectra

Reply:

PXRD patterns and IR spectra have been moved from the SI to the main manuscript (Figures 5 (Line between 433-434) and 6 (Line between 457-458)).

Comment:

2.2.9. Dissolution/permeation setup in a side-by-side cell; Why did the authors choose pH 6.8? I understand it is intestinal pH, is there any concept behind this?

Reply:

The medium pH 6.8 was chosen in order to maintain the same conditions for the solubility and dissolution/permeation experiments. As it has been indicated in our previous study, the pKa value of bicalutamide equals 11.5. From this, the same ionization state of the bicalutamide molecules at pH 6.8 and pH 7.4 can be stated and, as follows, the independence of the permeation rate on the small (0.6 units) differences in pH can be assumed.

Comment:

2.2.11. Again, what is the concept behind  pH 6.8 in receiver compartment, why not pH 7.4 or close to blood pH

Reply:

We agree with the Reviewer that usually buffer pH 7.4 in the receiver compartment is utilized. But as it was reported by Youdim et al. (DDT 2003, 8(21), 997-1003) the pH conditions for the permeability evaluation are strongly required in case of acidic and basic compounds (not characteristic for bicalutamide).

Comment:

Why did authors use both sink and non sink conditions for permeation studies?

Reply:

Non-sink conditions were studied in the dissolution/permeation setup in order to evaluate simultaneously the dissolution and permeation behavior of the solid bicalutamide/Ac-b-CD complex (under the varying donor concentration) and to reveal how the specific features of the dissolution (including the presence of the precipitation inhibitors) impact the permeation flux at different time points in the course of the experiment.

As opposed, the sink conditions were utilized in order to obtain the value of the permeability coefficient (Papp) of bicalutamide at a constant donor concentration. It is impossible in the non-sink experiment since the donor concentration varies upon the dissolution of the sample. Based on the comparison of Papp value through the cellulose membrane with the one measured in our previous study across the PermeaPad barrier the impact of the lipophilic layer was estimated.

Comment:

Did the authors store the samples at 40C/75RH to notice the changes in morphology and solubility?

Reply:

Stability experiments were not carried out in this work.

Reviewer 4 Report

The manuscript describes novel acetylated β-cyclodextrin cyclodextrin solid formulation of bicalutamide and its improved permeability in-vitro. Even though the manuscript is scientifically sound and convincing, the following points need to be addressed and discussed for acceptance in the final form.

  1)      What is the commercial and pharmaceutical relevance of acetylated β-cyclodextrin? Why was CAPTISOL not considered, given its current relevance in pharmaceutical settings?

 2)      The preparation of ground mixture has been described as milling of physical mixture. To this end, why was lyophilized complex not considered for milling as in the complex forms in solution state prior to mechanical processing?

 3)      The Figure 2 doesn’t effectively justify the caption. The subsection a) and b) is visible, but 1-3 in different media doesn’t seem to be included. The different colors are for different temperatures.  

Author Response

Reply to Reviewer_4

The manuscript describes novel acetylated β-cyclodextrin cyclodextrin solid formulation of bicalutamide and its improved permeability in-vitro. Even though the manuscript is scientifically sound and convincing, the following points need to be addressed and discussed for acceptance in the final form.

Comment:

1) What is the commercial and pharmaceutical relevance of acetylated β-cyclodextrin? Why was CAPTISOL not considered, given its current relevance in pharmaceutical settings?

Reply:

In the selection of cyclodextrin for improving the bicalutamide solubility we were guided by the following considerations. First of all, it is known that a solubilization potential of a certain cyclodextrin towards a specific compound is determined by the CD and drug structure, size, charge and other properties. On the first stage of our work, we analyzed the literature sources and found the information on the CDs which have been already used for BCL solubilization (this information is described in the Introduction and Results and discussion sections): b-CD and HP-b-CD. On the other hand, based on our previous experience of using cyclodextrins for solubilizing of new antifungal compound (T.V. Volkova and G.L. Perlovich, Eur. J. Pharm. Sci. 154 (2020) 105531), methylated CD showed a better solubilization power as compared to HP-b-CD.

Taking into account the premises, methylated and acetylated β-cyclodextrins were chosen for the bicalutamide solubility improvement. In addition, it seemed interesting to trace the impact of acetyl-group as compared to methyl-one on the solubilization efficiency.

Of course, acetylated β-cyclodextrin is not so intensively used as HP-b-CD or CAPTISOL. The pharmaceutical relevance and applicability can be evaluated in (Hirayama et al. J. Pharm. Sci. 1999, 88(10), 970–975, Soleimani et al. Carbohydr. Polym. 252 (2021) 117229).

We agree with the Reviewer that CAPTISOL can be advantageous in solubilization and can be applied in our future studies.

Comment:

2) The preparation of ground mixture has been described as milling of physical mixture. To this end, why was lyophilized complex not considered for milling as in the complex forms in solution state prior to mechanical processing?

Reply:

The BCL/Ac-b-CD complex was prepared by dry milling (grinding) without addition of the solvent (not in solution). Preparation of the complex by liophilization was not the aim of the study. The physical mixture was prepared by mixing BCL with Ac-b-CD in 1:1 ratio by the use of a spatula. The description of the procedure has been rewritten for the clarity (Line 190-192).

Comment:

3) The Figure 2 doesn’t effectively justify the caption. The subsection a) and b) is visible, but 1-3 in different media doesn’t seem to be included. The different colors are for different temperatures.

Reply:

The caption of Figure 2 has been corrected (Line between 324 and 325).

Round 2

Reviewer 4 Report

The authors have satisfactorily addressed the points of concern. Hence, the manuscript is recommended for acceptance for publication.